# Synthetic ECG signal generation using generative neural networks

**Edmond Adib**[1]*, **Fatemeh Afghah**[2], **John J. Prevost**[1]

**1** Electrical and Computer Engineering Department, University of Texas at San Antonio (UTSA), San Antonio, Texas, United States of America, **2** Department of Electrical and Computer Engineering, Clemson University, Clemson, South Carolina, United States of America

* edmond.adib@utsa.edu

**Data availability statement:** The dataset underlying the results presented in the study are available from PhysioNet MIT BIH Arrhythmia Dataset (https://physionet.org/content/mitdb/1.0.0/).

## Abstract

Electrocardiogram (ECG) datasets tend to be highly imbalanced due to the scarcity of abnormal cases. Additionally, the use of real patients' ECGs is highly regulated due to privacy issues. Therefore, there is always a need for more ECG data, especially for the training of automatic diagnosis machine learning models, which perform better when trained on a balanced dataset. We studied the synthetic ECG generation capability of 5 different models from the generative adversarial network (GAN) family and compared their performances, the focus being only on *Normal* cardiac cycles. Dynamic Time Warping (DTW), Fréchet, and Euclidean distance functions were employed to quantitatively measure performance. Five different methods for evaluating generated beats were proposed and applied. We also proposed 3 new concepts (threshold, accepted beat and productivity rate) and employed them along with the aforementioned methods as a systematic way for comparison between models. The results show that all the tested models can, to an extent, successfully mass-generate acceptable heartbeats with high similarity in morphological features, and potentially all of them can be used to augment imbalanced datasets. However, visual inspections of generated beats favors BiLSTM-DC GAN and WGAN, as they produce statistically more acceptable beats. Also, with regards to *productivity rate*, the Classic GAN is superior with a 72% productivity rate. We also designed a simple experiment with the state-of-the-art classifier (ECGResNet34) to show empirically that the augmentation of the imbalanced dataset by synthetic ECG signals could improve the performance of classification significantly.

## Introduction

Cardiovascular diseases are one the major causes of death (for example, 31% of all deaths in 2016) [1]. Electrocardiogram (ECG) analysis is routine in any complete medical evaluation, mostly due to the fact that it is painless, noninvasive, and can easily reveal arrhythmia. The classification and diagnosis of arrhythmias is usually done by experts of the domain, which is time consuming and prone to human error. Therefore, automatic ECG analysis and diagnosis is of crucial importance. Classical supervised machine-learning shallow algorithms have been employed extensively for the classification of abnormalities in ECG [2–4]. Deep learning

**Funding:** This research was partially supported by the Open Cloud Institute (OCI) at UTSA. The work of Fatemeh Afghah is supported by the National Science Foundation under Grant Number 2213915. There was no additional external funding received for this study.

**Competing interests:** The authors have declared that no competing interests exist.

unsupervised algorithms have also been successfully used and reached state-of-the-art results, reducing or eliminating the need for external feature engineering [5].

One of the challenges in the application of ML algorithms on ECG is that their datasets are usually highly imbalanced (with regards to the number of samples per class), which causes automatic diagnosis models perform poorly on them [6]. Moreover, collected ECG data are sometimes noisy and accompanied with different types of artifacts, which may render some samples unusable or require preprocessing [7]. On the other hand, in spite of transfer learning, ML algorithms generally still require huge datasets for training. All these challenges suggest and justify the need for more synthetic ECG data and richer and larger artificially augmented and balanced datasets. Another issue that justifies the need for synthetic ECG beat generation is privacy — unlike synthetic data, the ECG data of real-patients contain personal information, and thus is considered highly sensitive. Because of this, their use, even for scientific and research purposes, is highly regulated [8]. To address all these issues, the generation of synthetic ECG signals has been the focus of many studies [9–14].

The main objective of this research is to assess the capability of 5 models from the GAN family in generating synthetic *Normal* ECG heartbeats. Additionally, we present 5 different methods to systematically evaluate the performance of the models in generating synthetic ECG beats. To this end, three similarity measures were incorporated: Dynamic Time Warping (DTW), Fréchet, and Euclidean distance functions. This study is different from previous works [10,11,14] in that: (1) we employed more models from the GAN family, (2) we incorporated WGAN, (3) we used lead-I from the two, ([11] used lead-II), and (4) we present a systematic way to evaluate the performance of the models in generating synthetic data. In addition, we introduce three new concepts: *threshold, acceptable beat*, and *productivity rate*. Thresholds are used to mathematically define "acceptable beats" as well as screen the generated beats for those that are low quality. We suggest a way to compute the threshold as well and believe the productivity rate is a key indicator in performance evaluation. To the best of our knowledge, this is the first time such a systematic way of comparison has been presented. Also, we designed a binary classification experiment and showed that the augmentation of imbalanced datasets with synthetically generated ECG beats can improve the performance of classification comparably with the all real balanced dataset.

## Related works

Hong et al. [15] presented a comprehensive review and summary of the existing deep learning methods as well as challenges and opportunities in ECG analysis.

Delaney et al. [11] developed a range of GAN architectures to synthetically generate ECG beats. They used two evaluation metrics, *Maximum Mean Discrepancy* (MMD) and DTW, to quantitatively evaluate the generated beats and their suitability for real-world applications and used the Euclidean distance function in their "privacy disclosure test".

Hyland et al. [16] used two-layer LSTM architectures in both the generator and discriminator to generate synthetic ECG beats. For the evaluation of the performance of their models, they used MMD plus two innovative methods.

Zhu et al. [14] proposed a novel BiLSTM-CNN GAN to generate ECG beats and reported better performance compared to other existing models.

Wang et al. [10] used a 14-layer ACGAN to generate synthetic ECG beats for data augmentation. For the evaluation process, they used Euclidean, Pearson Correlation Coefficient (PCC), and Kulblack-Leibler similarity measures.

Zhang et al. [13] proposed a GAN model, whose generator was comprised of a 2 dimensional BiLSTM plus a CNN layer. In the discriminator, they used CNN and FC layers. They

used the standard 12 lead ECG signals, and studied four classes of arrhythmia, employing monoclass GAN models.

Wulan et al. [12] used three different GAN based models to generate 3 classes of ECG heart beats: Normal, Left Bundle Branch Block beat, and Right Bundle Branch Block beat. For evaluation, they used SVM (Support Vector Machine) classifier and GAN-train and GAN-test scores.

A comparison between some major works and our study is given in Tables 1, 2 and 3.

| | | | | | |
|---|---|---|---|---|---|
| 1 | MLII | 2 | This Study | 3 | Minibatch Discrimination |
| 4 | Maximum Mean Discrepancy | 5 | Batch Normalization | 6 | Dropout |
| 7 | Pearson Correlation Coefficient | 8 | Not Mentioned | 9 | Percent Root Mean Square Difference |
| 10 | Fréchet Distance | 11 | Interquartile Range | 12 | Skewness |
| 13 | Kurtosis | 14 | Fixed Window Segmentation | 15 | Instance Normalization |
| 16 | Support Vector Machine | 17 | GAN-train/GAN-test Score | | |

# Materials and mathematical background

## Generative models

Generative neural network models are powerful tools for learning the true underlying distribution of any kind of dataset in unsupervised settings. Two of the most commonly used families of generative models are *(Variational) Autoencoder-Decoder* (AE/VAE) and *Generative Adversarial Networks* (GAN).

**Table 1. Comparison with major related works - I**

| Ref. | Year | Main Objective | GAN Variant | Architecture (Gen. - Discr.) |
|---|---|---|---|---|
| [11] | 2019 | Generating Realistic Synthetic ECG Signal | Regular | LSTM-4CNN, BiLSTM-4CNN |
| [10] | 2019 | Dataset Augmentation and Balancing | ACGAN | 14CNN-16CNN |
| [14] | 2019 | Generating Realistic Synthetic ECG Signal | Regular | BiLSTM-(2CNN+FC) |
| [13] | 2021 | Fully Automated Synthetic ECG Generation | Regular | 2D BiLSTM 5CNN-2D 4CNN FC |
| [12] | 2020 | Generating Realistic Synthetic ECG Signal | DCGAN (SpectroGAN) Regular (WaveletGAN) | 2D 4TrCNN-2D 4CNN (SpectroGAN) 2D 3FC-2D 3FC (WaveletGAN) |
| ThS[2] | 2021 | Generating Realistic Synthetic ECG Signal | Regular, WGAN) | FC-FC, DC-DC, BiLSTM-DC, AE/VAE-FC, DC-DC (WGAN) |

**Table 2. Comparison with major related works - II.**

| Ref. | Dataset | Multiclass (study/model) | Mode Collapse Prevention | Metrics | Pre- processing |
|---|---|---|---|---|---|
| [11] | MIT-BIH (Lead II) | No/No (only Normal) | MBD [3] (didn't work) | MMD [4], DTW | Concat. of beats |
| [10] | MIT-BIH (Lead II) | Yes/Yes | BN [5], DO [6] (in Discr.) | ED, PCC [7], KL Div. (used templates) | NM |
| [14] | MIT-BIH (one lead) | [bc]NM [8] | DO | PRD [9], RMS, FD [10] | NM |
| [13] | 12 lead, PTB-XL, CCDD, CSE, Chapman, Private Domain | Yes/No | NM | MMD (IQR [11] SK [12] KU [13] (between train, test and synthetic sets) | FWS [14] |
| [12] | MIT-BIH (Lead I[1]) | Yes/Yes | IN [15] | SVM [16], GTrTs[17] | 4 second segmentation |
| ThS | MIT-BIH (Lead I) | No/No | BN, Visual Inspection | Original Methods | Pan-Tompkins |

**Table 3. Comparison with major related works - III.**

| Ref. | Batch Size | Optimization | Learning Rate | Hyper-Parameter Fine-Tuning | No. of Epochs |
|------|-----------|--------------|---------------|----------------------------|---------------|
| [11] | NM | Adam | NM | NM | 60 |
| [10] | NM | Adam | 0.0001 (G) 0.0002 (D) | NM | 150 |
| [14] | NM | NM | NM | NM | NM |
| [13] | 32 | NM | NM | NM | max 1000 (10 min.) |
| [12] | NM | RMSProp | 0.0001 (SectroGAN) 0.00015 (WaveletGAN) | NM | NM |
| ThS | 9 | Adam | 0.0002 | Used Recommended Suggestions | 30 |

**Autoencoder-decoders.** Through the assumption that data have been originally generated by a much lower-dimension latent variable space ($Z$), Autoencoder-Decoder (AE) models learn the distribution of the latent space and map from $Z$ (latent space) to $X$ (real data space) [17].

**Variational autoencoders.** In Variational Autoencoder-Decoder (VAE) networks, the bottleneck will be a *distribution* rather than a reduced dimension vector.

If $X$ is the input, $Z$ the latent random variables, $Q(Z|X)$ the encoder, and $P(X|Z)$ the decoder, then the objective function of VAE can be summarized as:

$$logP(X) - D_{KL}[Q(Z|X) \parallel P(Z|X)] = E[logP(X)] - D_{KL}[Q(Z|X) \parallel P(Z|X)] \quad (1)$$

The objective function can be interpreted as follows: maximizing the expectation of the input data while minimizing the KL distance (Kullback-Leibler Divergence, $D_{KL}$) between the encoder and decoder [17].

In order to make back-propagation feasible, a technique called the *Re-Parameterization Trick*, [17], is used in VAE, in which a *mean* vector along with a *standard deviation* vector are generated instead.

**Adversarial networks.** GAN architectures consist of two blocks of networks: the generator and the discriminator [18]. The generator, G, takes an input random vector $z$ and maps it into the data space (referred to as fake/synthesized data). The generator aims to fool the discriminator, i.e. make the discriminator mistakenly classify it as real.

The discriminator takes in an input (either fake/synthesized or real) and outputs a number between zero and one, representing the probability that the input is fake/synthesized or real, represented as $D(.) \in [0, 1]$ correspondingly.

The generator and the discriminator play a two-player zero-sum minimax game whose value function, $V(G, D)$, is defined as below [18]:

$$\min_G \max_D V(G, D) = E_{x \sim P_{data}(X)}[logD(X)] + E_{z \sim P_z(z)}[log(1 - D(G(z)))] \quad (2)$$

The GAN model implicitly finds the underlying distribution of the real data without any linkage or traceability between the generated data and the real data, which is required due to privacy concerns. Thus, the synthesized data by the generator have the same distribution as real data and can be used to enrich a dataset or to balance an imbalanced dataset. GAN models are a family and each member is named differently depending on the architecture

used in the generator and the discriminator. Examples of this include classic, LSTM (Long Short-Term Memory), DC (Deep Convolutional), et cetera.

## Experimental setup

### Dataset and segmentation

The MIT-BIH Arrhythmia [19,20] dataset is one of the most common benchmarks for ECG signal analysis and is used in this study as well. This dataset includes 48 30-minute two-channel ambulatory ECG records from 47 subjects studied by the BIH Arrhythmia Laboratory between 1975 and 1979. The recordings were digitized at 360 samples per second per channel with 11-bit resolution over a 10 mV range. This dataset is fully annotated with both beat-level and rhythm-level diagnoses. When segmented, the dataset is comprised of $109,338$ individual beats, of which $90,502$ beats are in the Normal class.

The MIT-BIH dataset is highly imbalanced and the beats are divided into 5 main classes: *N: Normal* (82.8%), *V: Premature ventricular contraction* (6.6%), *F: Fusion of ventricular and normal beat* (0.7%), *S: Supraventricular premature beat* (2.5%), and *Q: Unclassifiable beat* (7.3%). The class Q is not in fact a class per se, because any heartbeat that could not be classified has been put in this class; therefore, beats in this class do not follow a pattern as is the case in the other classes. One sample from the Normal class is shown in the Fig 6 (b). As the main objective in this study is to compare the capabilities of models in generating synthetic ECG beats, we focused on generating only one class of beats: *N*, which can be generalized to other classes.

We borrowed segmented dataset from Mousavi et al. [6], who used the Pan-Tompkins method [21] for segmentation. Their segmented beats are of the uniform length of 280, which were resampled to 256 in this study using the *scipy.signal.resample* function:

$$\mathcal{V} = \{\boldsymbol{v_i^k}\} \quad i = 1, \cdots, N_V^k \quad k = 1, \cdots, K \tag{3}$$

$$\boldsymbol{v_i^k} = [v_{i,1}^k, \ldots, v_{i,256}^k] \tag{4}$$

where $k$ is the class and $N_V^k$ is the number of the beats in class $k$. The dataset was filtered and only the Normal beat class was kept, so $k = 1$ and it is dropped hereafter. There are $N_V = 90,502$ individual beats ($\boldsymbol{v_i}$) in the filtered dataset space ($\mathcal{V}$).

### Model designs

A total of five models were utilized in this experiment, each of which is identified by a two digit code (01 to 05) for ease of reference. The details of the architectures of the models are shown in Tables 4, 5, 6, 7, and 8.

**Hyperparameter settings.** In all the models, the number of epochs is 30 with a batch size of 9. The optimizer used is ADAM with $\beta_1$ and $\beta_2$ equal to 0.5 and 0.999, respectively. The latent variable of a dimension of 100 is used at the generator input. Binary cross-entropy is used as the loss function. Gaussian Normal Distribution with $\mu$ of 0 and $\sigma$ of 0.02 is used for parameter initialization. In model 04 which is a hybrid model of VAE and GAN, the loss generator function is a weighted sum of the adversarial loss and the $L1$ loss between the decoded beats and real beats.

**Graphical representations of architectures.** Graphical representation of models 01 to 05 are shown in Figs 1, 2, 3, 4, and 5 respectively.

**Table 4. Classic GAN (01).**

| Layer | Generator | Discriminator |
|---|---|---|
| 1 | FC(100x128), L-ReLU(0.2) | FC(256x512), L-ReLU(0.2) |
| 2 | FC(128x256), BN, L-ReLU(0.2) | FC(512, 256), L-ReLU(0.2) |
| 3 | FC(256x512), BN, L-ReLU(0.2) | FC(256,1), sigmoid |
| 4 | FC(512, 1024), BN, L-ReLU(0.2) | - |
| 5 | FC(1024, 256), tanh | - |

**Table 5. DC-DC GAN (02).**

| Layer | Generator | Discriminator |
|---|---|---|
| 1 | ConvTr1d(100x512), BN, L-ReLU(0.2) | Conv1d(1, 64), L-ReLU(0.2) |
| 2 | ConvTr1d(512, 256), BN, L-ReLU(0.2) | Conv1d(64, 128), BN, L-ReLU(0.2) |
| 3 | ConvTr1d(256, 128), BN, L-ReLU(0.2) | Conv1d(128, 256), BN, L-ReLU(0.2) |
| 4 | ConvTr1d(128, 64), BN, L-ReLU(0.2) | Conv1d(256, 512), BN, L-ReLU(0.2) |
| 5 | ConvTr1d(64, 1), BN, L-ReLU(0.2) | Conv1d(512, 1), FC(13, 1), sigmoid |
| 6 | FC (64, 256), tanh | - |

**Table 6. BiLSTM-DC GAN (03).**

| Layer | Generator | Discriminator |
|---|---|---|
| 1 | BiLSTM(100, 1000), 2 layers, | Conv1d(1, 64), L-ReLU(0.2) |
| 2 | FC(1000*2, 256), tanh | Conv1d(64, 128), BN, L-ReLU(0.2) |
| 3 | - | Conv1d(128, 256), BN, L-ReLU(0.2) |
| 4 | - | Conv1d(256, 512), BN, L-ReLU(0.2) |
| 5 | - | Conv1d(512, 1), FC(13, 1), sigmoid |

**Table 7. AE/VAE-DC GAN (04).**

| Layer | Encoder | Decoder | Discriminator |
|---|---|---|---|
| 1 | FC(256, 512), L-ReLU(0.2) | FC(10, 512), L-ReLU(0.2) | FC(10, 512), L-ReLU(0.2) |
| 2 | FC(512, 512), BN, LReLU | FC(512, 512), BN, L-ReLU(0.2) | FC(512, 256), L-ReLU(0.2) |
| 3 (mu) | FC(512, 10) | FC(512, 256), tanh() | FC(256, 1), sigmoid |
| 4 (logvar) | FC(512, 10)) | - | - |
| 5 (Output layer) | Reparameterization (mu, logvar) | - | - |

**Table 8. WGAN (05).**

| Layer | Generator | Discriminator |
|---|---|---|
| 1 | ConvTr1d(100x2048), BN, ReLU | Conv1d(1, 64), L-ReLU(0.2) |
| 2 | ConvTr1d(2048, 1024), BN, ReLU | Conv1d(64, 128), BN, L-ReLU(0.2) |
| 3 | ConvTr1d(1024, 512), BN, ReLU | Conv1d(128, 256), BN, L-ReLU(0.2) |
| 4 | ConvTr1d(512, 256), BN, ReLU | Conv1d(256, 512), BN, L-ReLU(0.2) |
| 5 | ConvTr1d(256, 128), BN, ReLU | Conv1d(512, 1024), BN, L-ReLU(0.2) |
| 6 | ConvTr1d(128, 64), BN, ReLU | Conv1d(1024, 2048), BN, L-ReLU(0.2) |
| 7 | Conv1d (64, 1), tanh | Conv1d(2048, 1) |

## Similarity measures (distance functions)

Currently, there is a lack of consensus on the best evaluation metric for the performance of the generative models [22] and researchers mostly resort to the subjective expert-eye evaluation. In general, a distant function *DF* has a scalar output that quantifies the proximity (or distance) between its two input beats:

$$DF(\mathbf{x}, \mathbf{y}): \quad \mathcal{R}^{256} \text{ x } \mathcal{R}^{256} \to \mathcal{R} \tag{5}$$

**Dynamic time warping measure (DTW).** DTW belongs to a family of measures known as "elastic dissimilarity measures" and it works by optimally aligning (warping) the time scale in a way that the accumulated cost of this alignment is minimal [23]. It constructs a *cost matrix D* based on the two time-series being compared, *x* and *y*. The elements of matrix *D* are defined, by a recurrent formula:

$$D_{i,j} = f(x_i, y_j) + min\{D_{i-1,j}, D_{i,j-1}, D_{i-1,j-1}\}$$

For $i = 1, \dots, M$ and $j = 1, \dots, N$ where *M* and *N* are the lengths of the two time series. The local cost function $f(.,.)$, also called *sample dissimilarity function*, is usually the Euclidean distance. The final DTW value typically corresponds to the total accumulated cost, i.e. $d_{DTW}(x, y) = D_{M,N}$ [23].

**Fréchet distance measure.** If $P = (u_1, u_2, \dots, u_p)$ and $Q = (v_1, v_2, \dots, v_q)$ are two time series, a *coupling L* between *P* and *Q* is defined as the set of the links:

$$(u_{a_1}, v_{b_1}), (u_{a_2}, v_{b_2}), \dots, (u_{a_m}, v_{b_m}) \tag{6}$$

such that $a_1 = 1$, $b_1 = 1$, $a_m = p$ and $b_m = q$, and for all $i = 1, \dots, q$, $a_{i+1} = a_i$ or $a_{i+1} = a_i + 1$ and $b_{i+1} = b_i$ or $b_{i+1} = b_i + 1$. The length $\|L\|$ of the coupling *L* is defined as the longest (maximum Euclidean distance) in the link *L*:

$$\|L\| = \max_{i=1,\dots,m} d(u_{a_i}, v_{b_i}) \tag{7}$$

*Model 01*

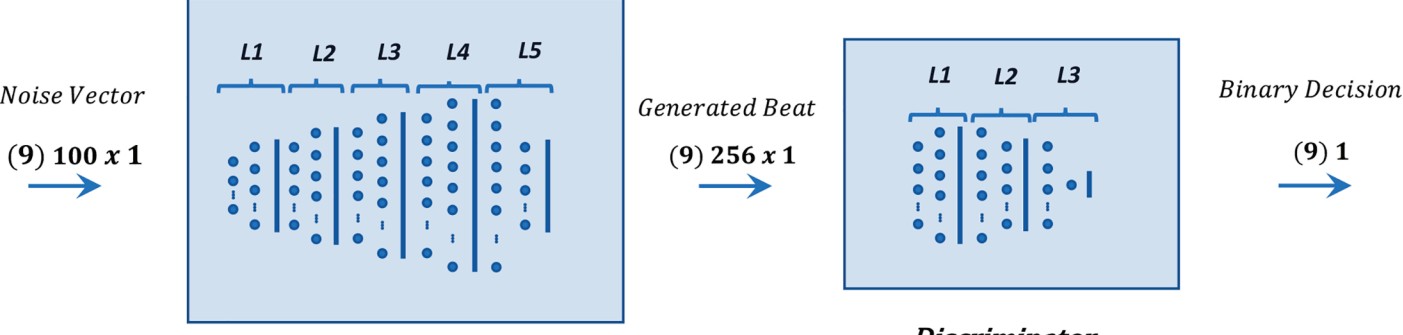

**Fig 1. Graphical representation of model 01.**

*Model: 02*

$$z = (z_1, z_2 \ldots, z_{100}), \; z_i \sim \mathcal{N}(0, 1)$$

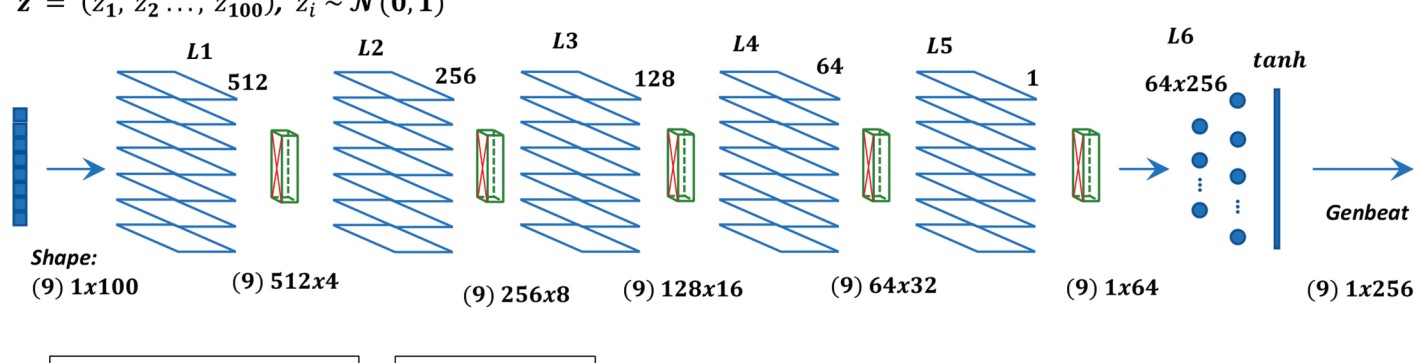

*Model: 02*

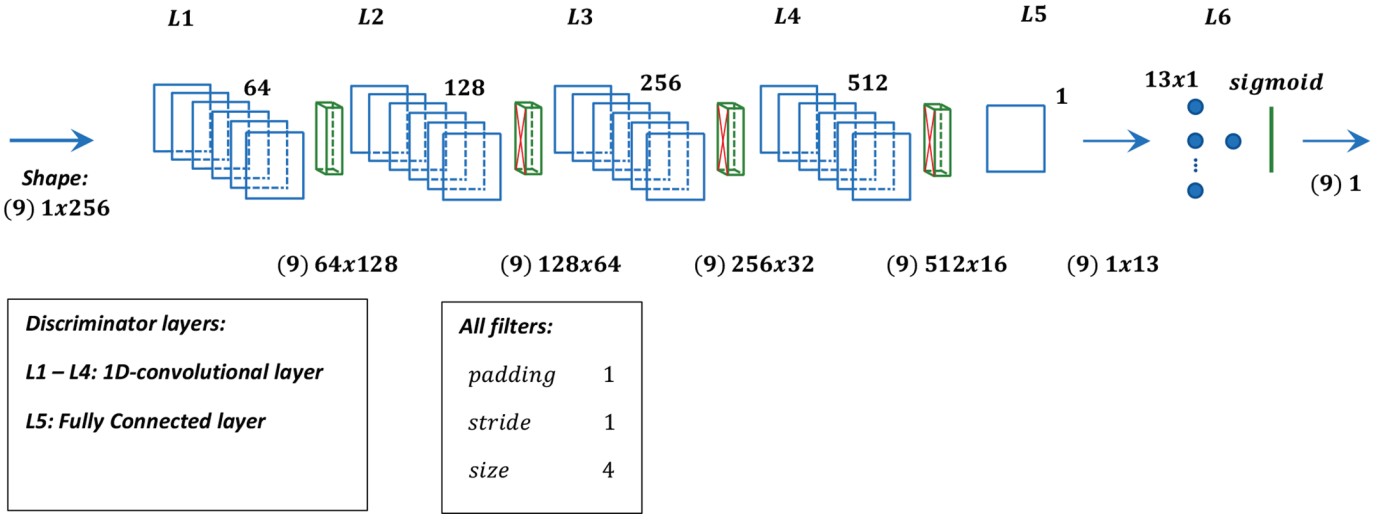

**Fig 2. Graphical representation of model 02.**

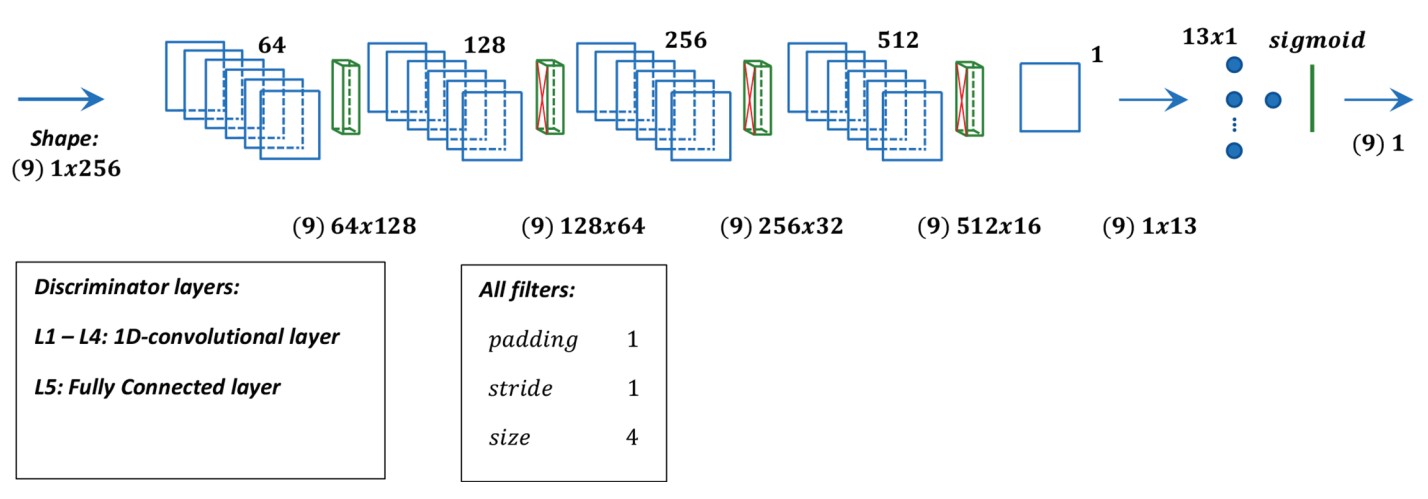

**Fig 3. Graphical representation of model 3.**

**Model 04**

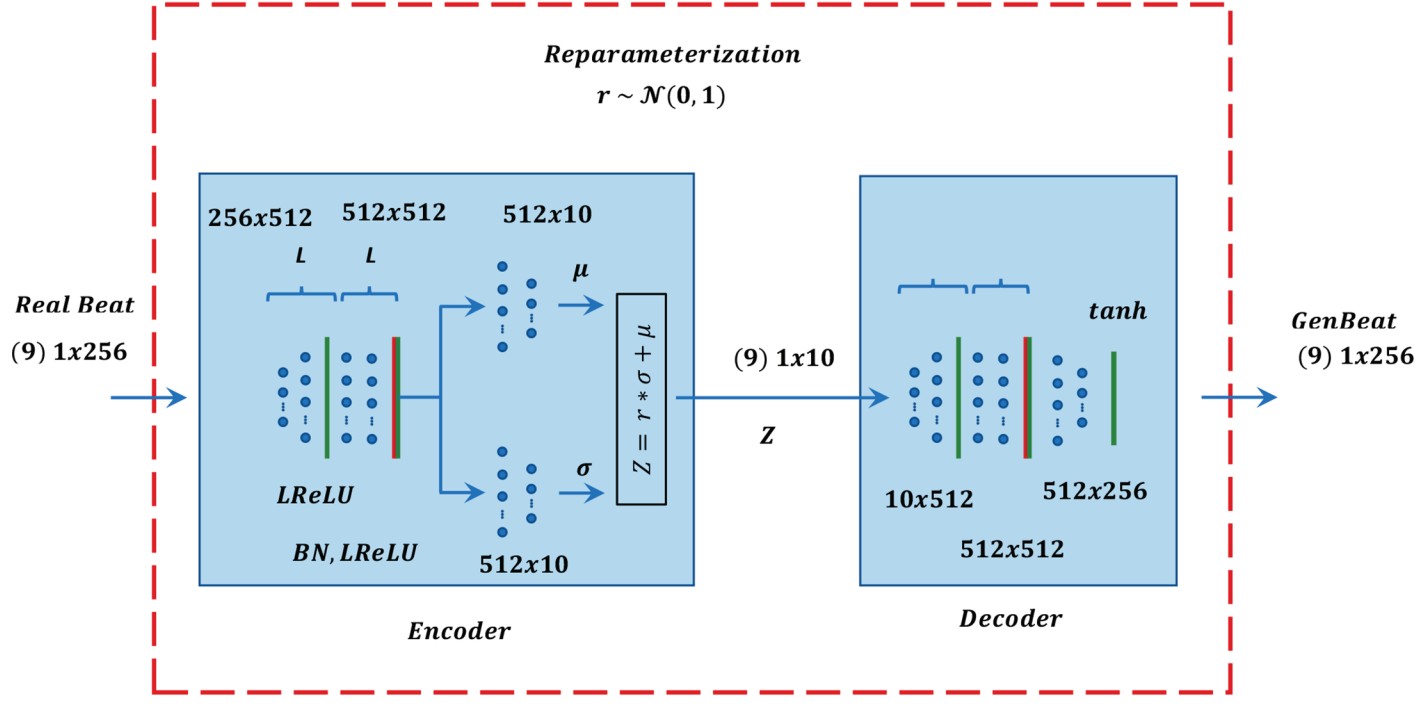

**model: 04**

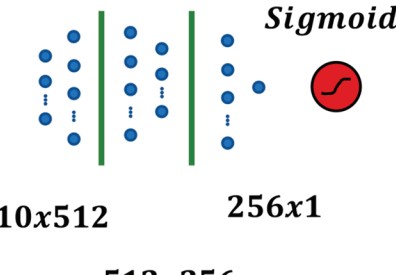

**Discriminator**

**Fig 4. Graphical representation of model 4.**

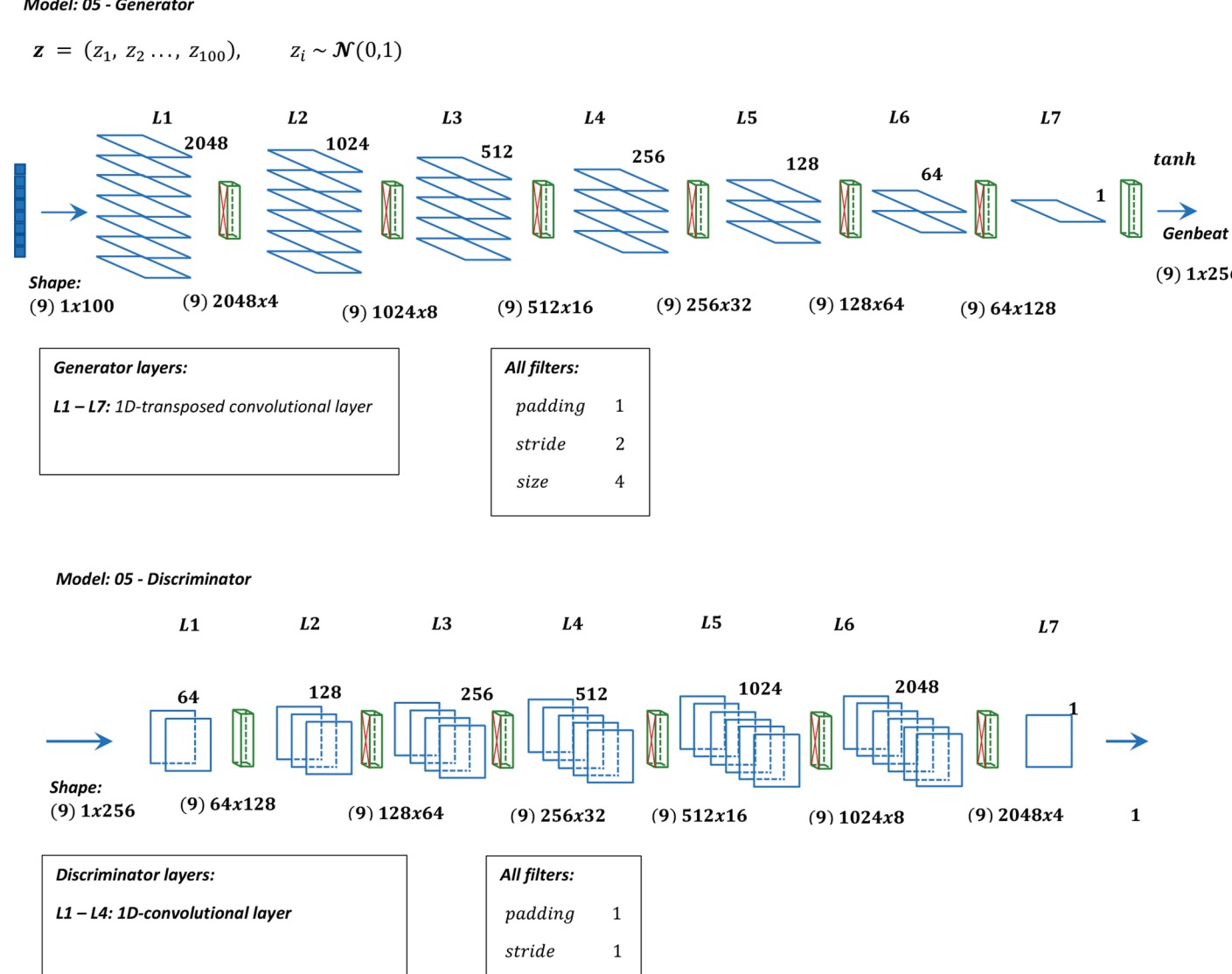

**Fig 5. Graphical representation of model 5.**

then the Fréchet distance between $P$ and $Q$ is defined as [24]:

$$\delta_{dF} = min\{\|L\| \mid L \text{ is a coupling between } P \text{ and } Q\} \tag{8}$$

**Euclidean distance measure.** The Euclidean distance between two time series, $P = (u_1, u_2, \ldots, u_p)$ and $Q = (v_1, v_2, \ldots, v_q)$ is defined as:

$$d(P, Q) = \sqrt{(u_1 - v_1)^2 + \cdots + (u_n - v_n)^2} \tag{9}$$

Fréchet distance function fulfills all the properties required by metric spaces (e.g., commutative, triangle property, …) and can be used as a *metric*. However, DTW and Euclidean

measures do not satisfy the triangle property and are not a metric as required by metric spaces.

## Templates

For each class, there is one template which is the quintessential time-series of that class and distinctly represents all the morphological features and patterns of the class. Distance functions take the template as well as a generated beat as inputs and generate a scalar number, which signifies the proximity of the two time-series. The following two approaches are available for developing/selecting templates.

**Statistically-Averaged Beat (SAB) approach.** Since all beats have an equal number of time steps (256), it is sensible that the template is defined as some sort of "mean of the class" such that the value at each time step is computed as the mean across all the beats of the class at that time step:

$$\bar{v}_j = \frac{1}{N_V} \sum_{i=1}^{N_V} v_{i,j} \quad j = 1, \dots, 256 \tag{10}$$

$$t = [\bar{v}_1, \dots, \bar{v}_{256}] \tag{11}$$

where $N_V$ is the number of beats in the set and $t$ is the template. One sample of SAB template is shown in Fig 6(a).

**Expert-eye/random approach.** In this approach, the original dataset is visually inspected by a domain expert to find the "most fit sample" that meets all the morphological characteristics of that class. In this experiment, the expert-eye approach is employed to select the template, which is shown in Fig 6(b).

## Evaluating the generated beats and comparison between models

Evaluating the quality of the generated beats and the comparison between the performances of the models can be accomplished through one of the following four methods.

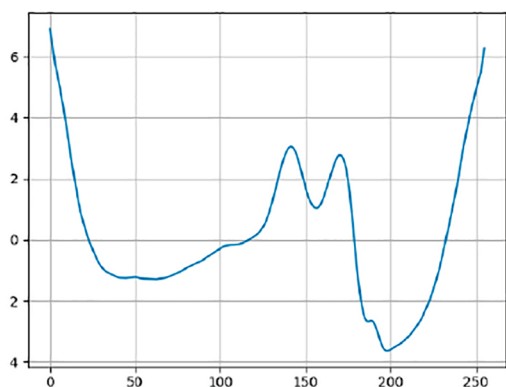

$(a)$ Template, Statistically Averaged Beat

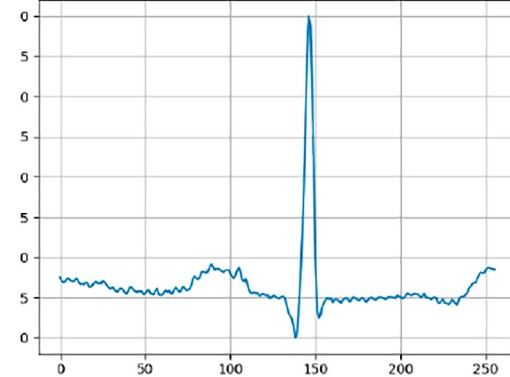

$(b)$ Template, Randomly Selected (used in this study)

**Fig 6. Templates.**

**Method 1.** To assure the proximity of the two sets, the *whole set* of generated beats should be cross-compared to the *whole original dataset*, element by element. The outcome of this analysis (i.e., the mean distance) is a deterministic single number (with no randomosity) representing the average distance between the two sets. If $\mathcal{V}$ is the dataset space with $N_V$ elements in it and $\mathcal{G}$ is the generated beats space, i.e.:

$$\mathcal{G} = \{\boldsymbol{g_i}\} \qquad i = 1, ..., N_G \tag{12}$$

$$\boldsymbol{g_i} = [g_{i,1}, ..., g_{i,256}], \tag{13}$$

then the average distance between the two sets, $d_{ave}^{DF}$, is:

$$s_1^{DF} = d_{ave}^{DF} = \frac{1}{N_V N_G} \sum_{i=1}^{N_V} \sum_{j=1}^{N_G} DF(\boldsymbol{v_i}, \boldsymbol{g_j}) \tag{14}$$

However, this analysis usually is not pragmatic, as it requires tremendous computational power as the number of elements in the sets increases. To approximate, for cross comparison, one can instead apply Eq 14 on two randomly selected portions from the two sets with sizes $N_V^*$ and $N_G^*$. Of course, the method of sampling of the portions $N_V^*$ and $N_G^*$ plays a significant role in the outcome and makes this process stochastic. The size of the portions depends on the available computational power (the more there is, the more accurate the results will be). In this experiment, we used $N_G^* = 300$ generated beats from each model (10 beats from each of the 30 epochs) and cross-compared them against $N_V^* = 300$ randomly selected beats from the original dataset. Obviously, this process is stochastic and the outcome depends on the particular portions selected (Table 9).

**Method 2.** In this approach, the template is randomly selected from the original dataset and all the generated beats are compared with it. The average distance of all generated beats from the template is the score for that model:

$$s_2^{DF} = \frac{1}{N_G} \sum_{i=1}^{N_G} DF(\boldsymbol{v_i}, \boldsymbol{t}) \tag{15}$$

This method is also obviously *stochastic*, as the outcome depends on the initial choice of the template. However, there is a constraint on the selected template which must have all the morphological features required by the class. Therefore, the variation is very limited and the results are more reliable. To select the best model, the scores are compared with each other in Table 10.

**Method 3.** In this method, a template is randomly selected from the original dataset as in Method 2. Then, all the beats generated by each model are measured against the template and the beat which has produced the *minimum distance function value* is reported as the "best

**Table 9. Method 1 (portions of the two sets compared).**

| Model | Model | $s_1^{DTW}$ | $s_1^{Fré}$ | $s_1^{Euc}$ |
|---|---|---|---|---|
| 01 | *Classic GAN* | *3.953* | *0.589* | 8.325 |
| 02 | *DC-DC GAN* | 5.313 | 0.862 | 9.390 |
| 03 | *BiLSTM-DC GAN* | 4.535 | 0.625 | 8.557 |
| 04 | *AE/VAE-DC GAN* | 4.357 | 0.622 | *8.230* |
| 05 | *WGAN* | 4.401 | 0.681 | 8.486 |

**Table 10. Method 2 (all generated beats compared with one randomly selected template, averages).**

| Model | Model | $s_2^{DTW}$ | $s_2^{Fré}$ | $s_2^{Euc}$ |
|-------|-------|-------------|-------------|-------------|
| 01 | *Classic GAN* | *4.13* | 0.595 | 8.44 |
| 02 | *DC-DC GAN* | 5.66 | 0.863 | 9.75 |
| 03 | *BiLSTM-DC GAN* | 4.33 | *0.594* | *8.29* |
| 04 | *AE/VAE-DC GAN* | 4.52 | 0.627 | 8.34 |
| 05 | *WGAN* | 4.59 | 0.693 | 8.71 |

beat" for that model. The score of the model is the distance of the best beat of that model with the template. This method measures the ultimate power of each model in getting as close to the template as possible (Table 11):

$$v_{best}^{DF} = \underset{v_i \in \mathcal{V}}{\arg\min} \, DF(v_i, t) \tag{16}$$

$$s_3^{DF} = DF(v_{best}^{DF}, t) \tag{17}$$

**Method 4.** In this method, a *threshold* is defined for each similarity measure ($\eta^{DF}$). Any generated beat ($g_i$) with a distance function value less than the threshold is considered as an *acceptable beat*, with respect to that distance function, i.e.,:

$$\mathcal{G}^{acc,DF} = \{g_i : DF(g_i, t) \leq \eta^{DF} \quad i = 1, ..., N_G\} \tag{18}$$

$$N_G^{acc,DF} = n(\mathcal{G}^{acc,DF}) \tag{19}$$

where $n(.)$ is the number of the element in the set. Then, the *Productivity Rate* (i.e. the percentage of the acceptable beats among all the generated beats) is calculated as the discriminating factor between the models:

$$s_4^{DF} = Prod^{DF} = \frac{n(\mathcal{G}^{acc,DF})}{n(\mathcal{G})} = \frac{N_G^{acc,DF}}{N_G} \tag{20}$$

The model which produces the highest productivity rate is selected as the best in performance (Table 12). The value choice of the threshold is rather arbitrary, experience-based, and must be validated by a domain expert. It can be set at a factor of the minimum distance:

$$\eta^{DF} = a \, s_3^{DF} \qquad a \in \mathcal{R} \tag{21}$$

**Table 11. Method 3 (best generated beat - Minimum Distance Functions).**

| Model | Model | $s_3^{DTW}$ | $s_3^{Fré}$ | $s_3^{Euc}$ |
|-------|-------|-------------|-------------|-------------|
| 01 | *Classic GAN* | 0.510 | *0.0844* | 0.890 |
| 02 | *DC-DC GAN* | 0.505 | 0.120 | 1.38 |
| 03 | *BiLSTM-DC GAN* | 0.425 | 0.0966 | 3.42 |
| 04 | *AE/VAE-DC GAN* | 0.505 | 0.108 | 1.02 |
| 05 | *WGAN* | *0.311* | 0.0981 | *0.610* |

**Table 12. Method 4 (productivity rate - percent of acceptable beats).**

| Model | Model | $s_4^{DTW}$ | $s_4^{Fré}$ | $s_4^{Euc}$ |
|-------|-------|-------------|-------------|-------------|
| 01 | *Classic GAN* | *72.3* | *60.0* | 10.5 |
| 02 | *DC-DC GAN* | 26.8 | 18.6 | *11.2* |
| 03 | *BiLSTM-DC GAN* | 54.2 | 47.0 | 0.437 |
| 04 | *AE/VAE-DC GAN* | 49.7 | 37.7 | 9.80 |
| 05 | *WGAN* | 49.0 | 38.05 | 8.50 |

In this experiment, for each distance function, the threshold is essentially computed separately as the arithmetic mean between the *minimum* and the *average* of the values of that particular distance function among all the generated beats:

$$\eta^{DF} = \frac{s_3^{DF} + s_2^{DF}}{2} \tag{22}$$

**Method 5.** In this method, an expert with domain-specific knowledge looks at the entire set of generated beats and gives their subjective judgment on the performance of the model. This is accomplished by inspecting the existence of the morphological features of the beat class in the set of generated beats. This method can also be used simply to validate the other aforementioned methods.

## Efficacy of synthetic ECG augmentation

A simple experiment has been designed to show the efficacy of the augmentation. A subset of the MIT-BIH Arrhythmia dataset is selected which contained only two classes: *N* (Normal Sinus Beat) and *L* (Left Bundle Block Branch Block). Then, a state-of-the-art classifier (ECGResNet34) is trained on *(i)* a balanced binary real dataset (*L*: 6455 and *N*: 6457) and *(ii)* an *imbalanced* dataset created by sampling the original dataset (*L*: 6460 and *N*: 500) so that the classifier performs poorly on classification. And finally, *(iii)* the imbalanced dataset is *balanced* back again (*L*: 6458 and *N*: 6454) by augmenting with synthetically generated beats. The classifier is trained on each of these three training sets and the classification metrics and confusion matrices are compared with each other in all the three cases. The test set (unseen data) is the same in all three cases (*L*: 1607 and *N*: 1609). The classifier used is ResNet34 [25] which is a 34-layer model and is the state-of-the-art in classification of images (*2D*). It incorporates residual building blocks following the residual stream logic: $F(\boldsymbol{x}) + \boldsymbol{x}$. Each building block is comprised of two $3 \times 3$ convolutional layers where the residual stream, $\boldsymbol{x}$, goes directly from the input to the outlet of the block which prevents deterioration of training accuracy in deeper models [25]. This classifier is pretrained on the ImageNet dataset (more than $100,000$ images in 200 classes). We used its **1D** implementation [26] to classify the heartbeats.

## Platform

Two different machines have been utilized in this experiment: a Dell Alienware with Intel i9-9900k at 3.6 GHz (8 cores, 16 threads) microprocessor, 64 GB RAM, and NVIDIA GeForce RTX 2080 Ti graphics card with 24 GB RAM, and a personal Dell G7 laptop with an Intel i7-8750H at 2.2 GHz (6 cores, 12 threads) microprocessor, 20 GB of RAM, and NVIDIA GeForce 1060 MaxQ graphics card with 6 GB of RAM.

The codes were written in Python 3.8, and PyTorch 1.7.1 was used as the main deep learning network library, as it makes the migration between CPU and GPU as well as the

back-propagation and optimization much easier, thanks to its dynamic computational graph feature. The codes are available on the GitHub page of the paper (https://github.com/mah533/Synthetic-ECG-Generation—GAN-Models-Comparison).

## Results and discussion

### Templates and typical normal beat

A statistically-averaged beat (SAB) template is shown in Fig 6(a). The downside to SABs is that although all the beats have the same number of time-steps, small horizontal shifts of morphological features in the temporal axis are inevitable (for instance, as a result of the segmentation process or heart rate variability). Consequently, in the calculation of mean values ($\bar{v}_j$), more often than not, not all of the corresponding points are averaged together. Therefore, the generated template (Fig 6(a)) looks completely distorted and totally different from the Normal beat . In other words, the morphological characteristic features of that class are not visually distinguishable anymore. However, it should not be forgotten that on an average basis and from the statistical aspect, the SAB template is the best representation of information from all the samples in the class and it has been used in similar studies [10].

### Generated beats

Some of the generated beats by different models are presented in Figs 7, 8, 9, 10, and 11. Figures in columns (a), (b) and (c) are the generated beats with minimum DTW, Fréchet,

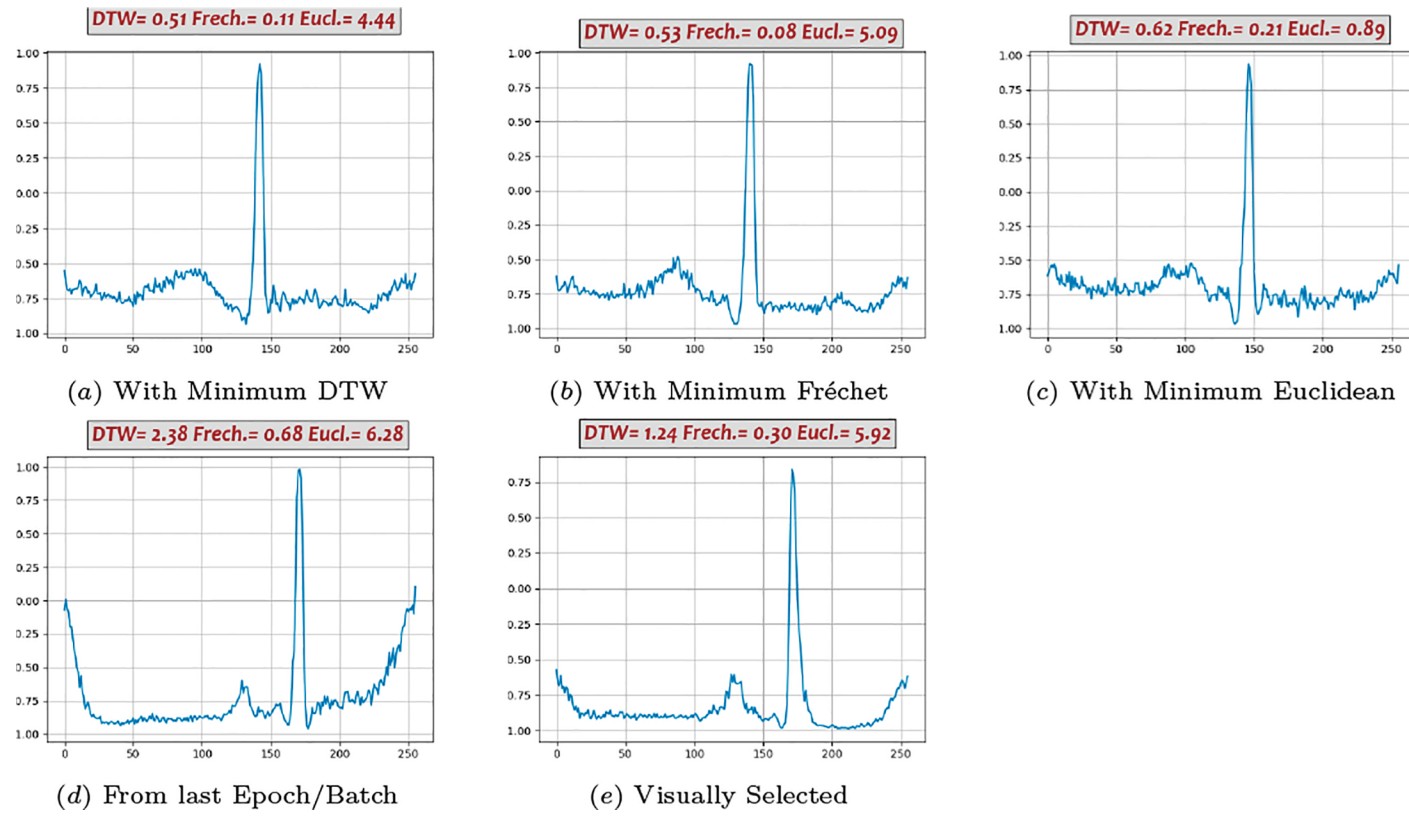

(a) With Minimum DTW

(b) With Minimum Fréchet

(c) With Minimum Euclidean

(d) From last Epoch/Batch

(e) Visually Selected

**Fig 7. Generated beats, Classic GAN (01).**

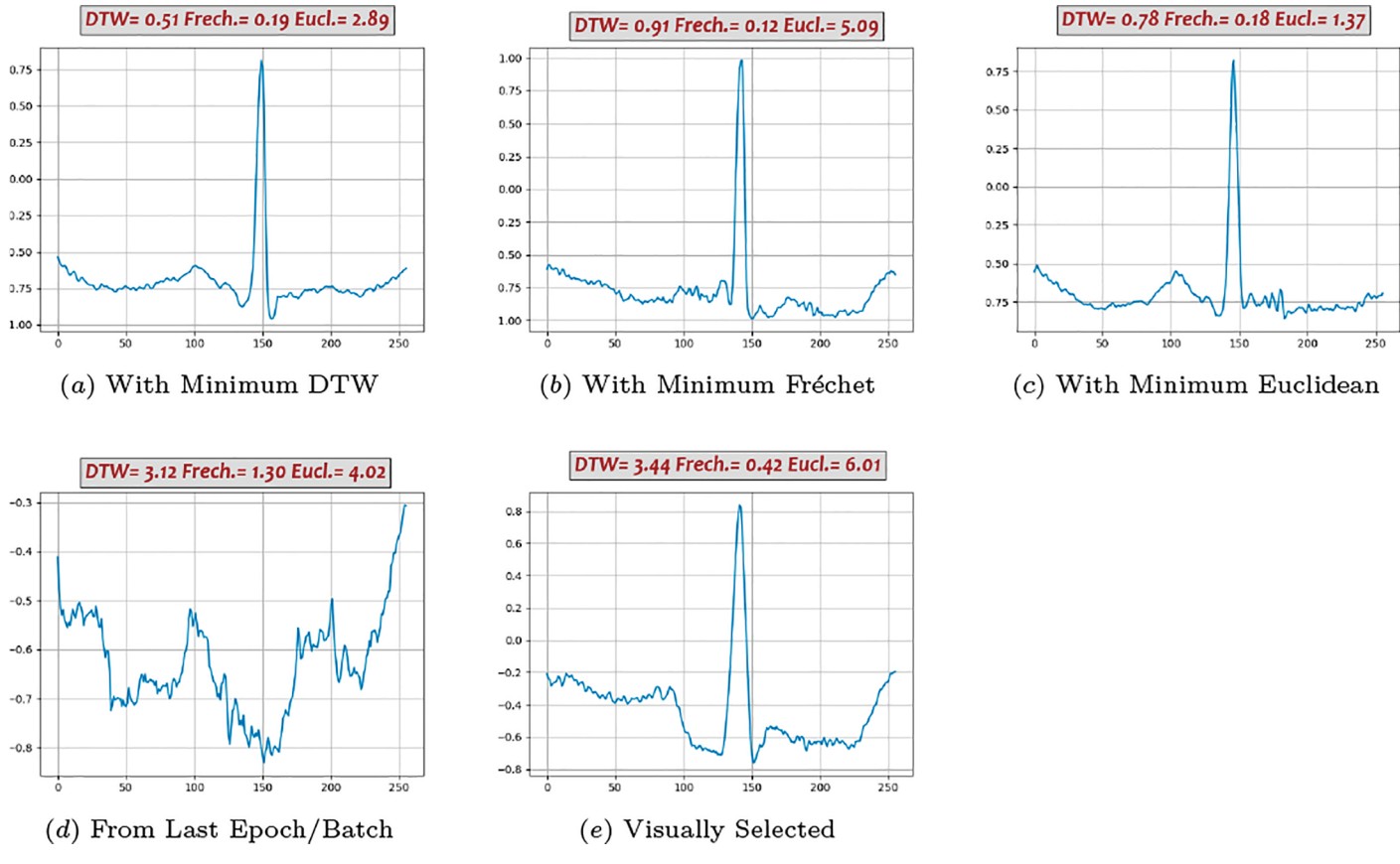

**Fig 8. Generated beats, DC-DC GAN (02).**

and Euclidean distance functions, (i.e., $v_{best}^{DTW}$, $v_{best}^{Fré}$ and $v_{best}^{Euc}$) respectively. The calculated values of all three distance functions are shown on each plot as well for comparison. In column (d), a beat selected from the last batch of the last iteration in the last epoch is shown, which in fact represents the *maximally trained models'* outputs. It can be seen that, after convergence, more training does not necessarily produce a better beat, neither in appearance nor in terms of quantitative proximity. Finally, in column (e), a beat that is visually close enough to the template in terms of morphological features is selected by ab expert.

## Distance and loss functions

The trends of all three similarity measures as well as the generator and discriminator loss functions against the epoch number are plotted and shown in Fig 12. All graphs of the DC-DC GAN model (02) suffer from severe fluctuations, which is a result of a convergence issue. Fluctuation exists in other models as well, but they are not as severe.

## Performance metrics

Table 9 summarizes the performance metric $s_1^{DF}$ (i.e., $s_1^{DTW}$, $s_2^{Fré}$ and $s_1^{Euc}$) of the five models. As shown here, Classic GAN (in terms of DTW and Fréchet distance functions) and AE/VAE-DC GAN (in terms of the Euclidean distance function) seemingly generate sets of beats closest to the original dataset. However, it should be noted here that this analysis is stochastic, as

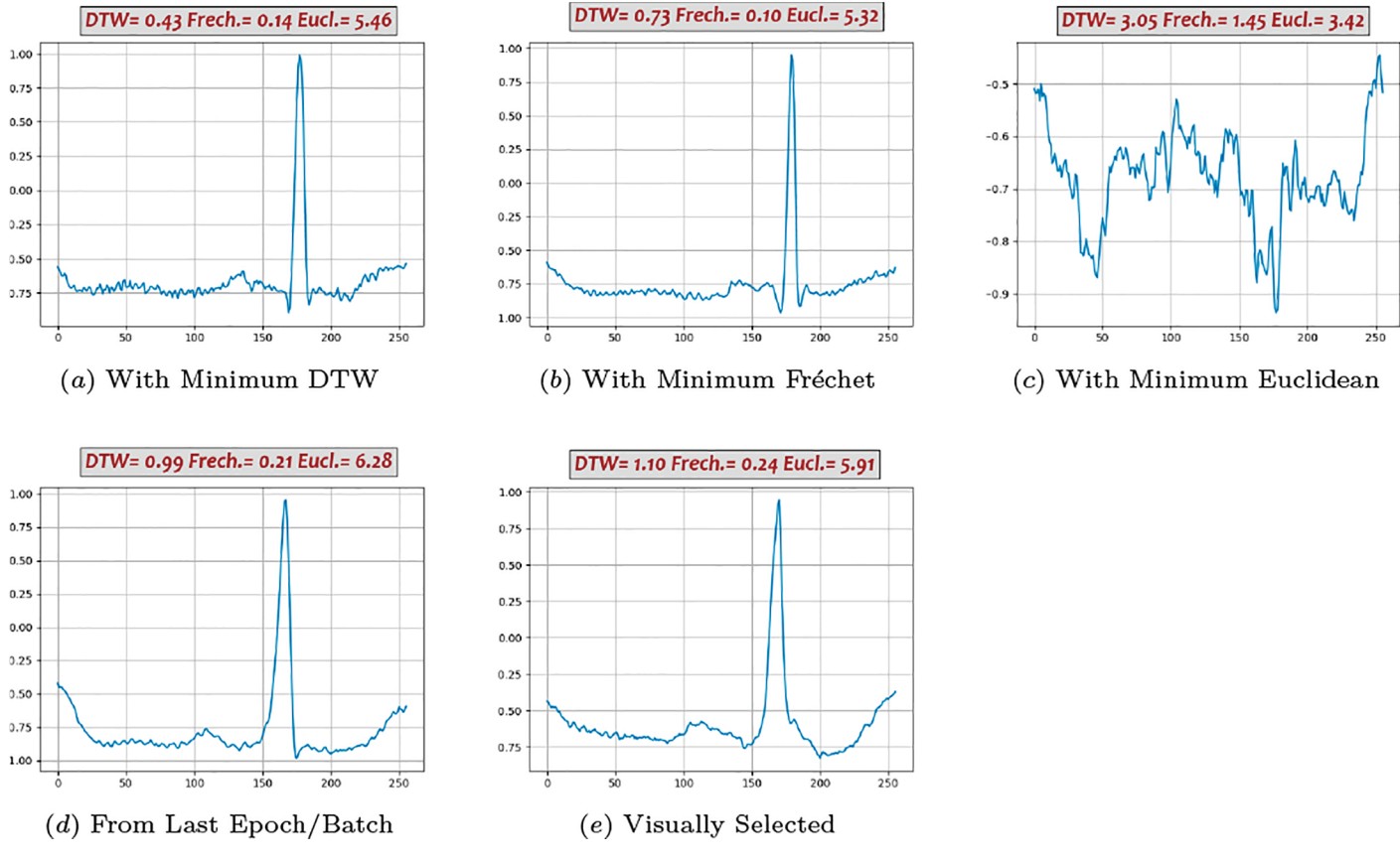

**Fig 9. Generated beats, BiLSTM-DC GAN (03).**

the outcome depends on the method of sampling, i.e. the way the *portions* are selected. Therefore, the resulting outcomes are basically just one realization of the corresponding random variables and consequently cannot be used for *deterministic* judgments. Nevertheless, these numbers show that all the models are, give or take, in the same ballpark range.

Table 10 shows the result of the comparison using Method 2 ($s_2^{DF}$, the average distances from the template). Similar to Method 1, this process is also random because it depends on the choice of template. However, since the selected template is constrained to have all the morphological features of the class, as any other qualified candidate is, the outcome numbers are much more reliable. The results show that with respect to the DTW distance function Classic GAN, with respect to Fréchet, both the Classic and BiLSTM-DC GAN models equally and, with respect to Euclidean distance function BiLSTM-DC GAN, perform the best.

The result of the analysis using Method 3, $s_3^{DF}$ (the distance of the best generated beat from the template), are shown in Table 11. The results show that in terms of DTW and Euclidean, WGAN perform the best, and in terms of Fréchet distance function, Classic GAN perform the best.

Assessment in terms of productivity rates ($s_4^{DF}$, Table 12), reveals that 72.3% and 60.0% of the generated beats by the Classic GAN (01) model are acceptable with respect to the DTW threshold ($\eta^{DTW}$) and Fréchet threshold, respectively.

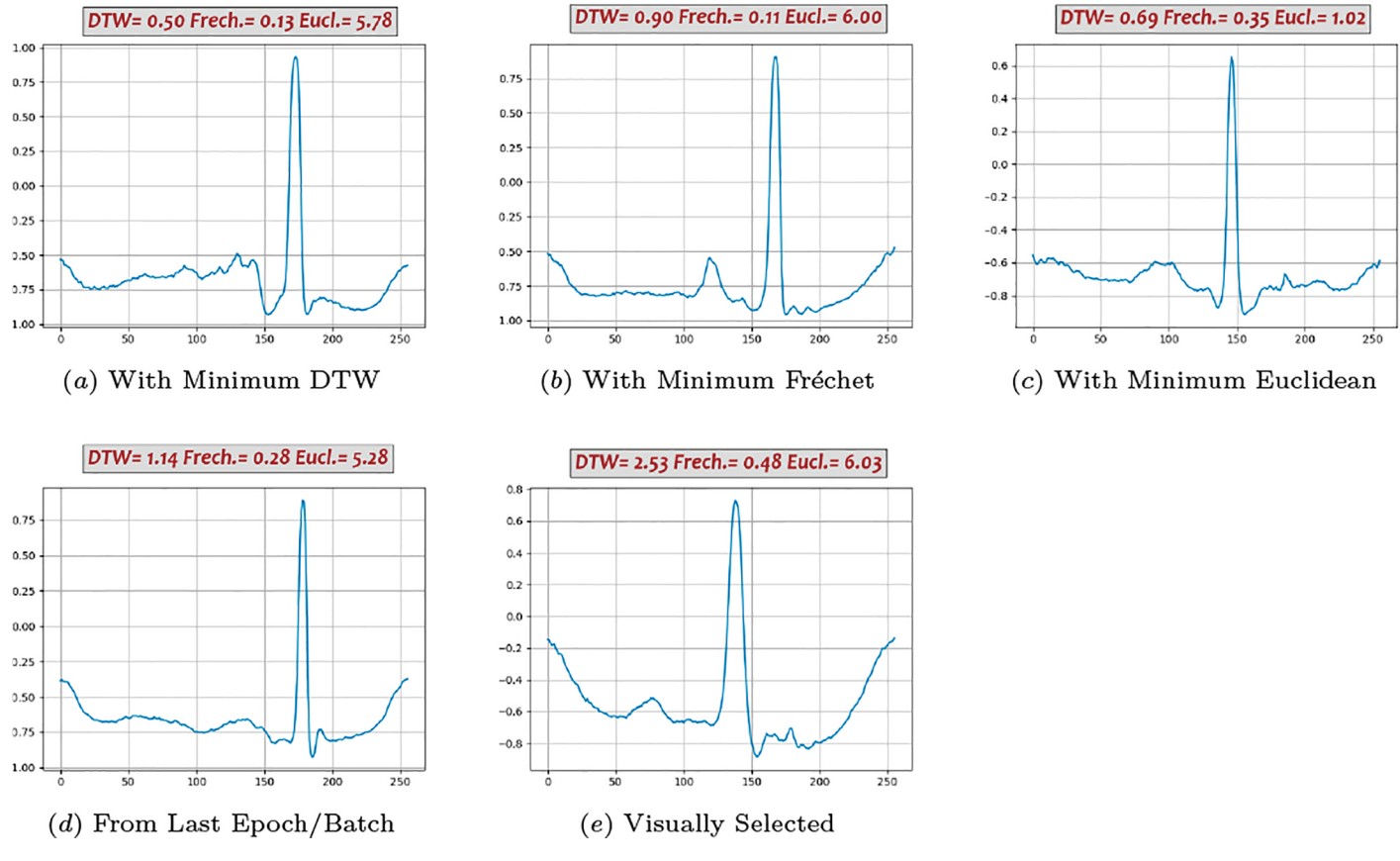

**Fig 10. Generated beats, AE/VAE GAN (04).**

The Euclidean measure selects the DC-DC GAN with only 11% of success. This method, like Method 2, is essentially random, but since the selected template is constrained, its randomosity is limited.

Visual inspection of the generated beats by a domain-expert knowledge (Method 5) suggests subjectively that WGAN and BiLSTM-DC GAN models produce more acceptable beats than the other models.

### Efficacy of augmentation

Comparing the results in Tables 15 and 14 shows that the augmentation of the imbalanced dataset with synthetically generated beats can improve the classification drastically, almost as in real balanced dataset Table 13. Same trend is noticeable from confusion matrices Tables 16a), 16b) and 16c).

## Conclusion

Machine Learning automatic ECG diagnosis models classify ECG signals based on morphological features. ECG datasets are usually highly imbalanced due to the fact that the anomaly cases are scarce compared to the abundant Normal cases. Additionally, because of privacy concerns, not all the collected data from real patients are available as training sets. Therefore,

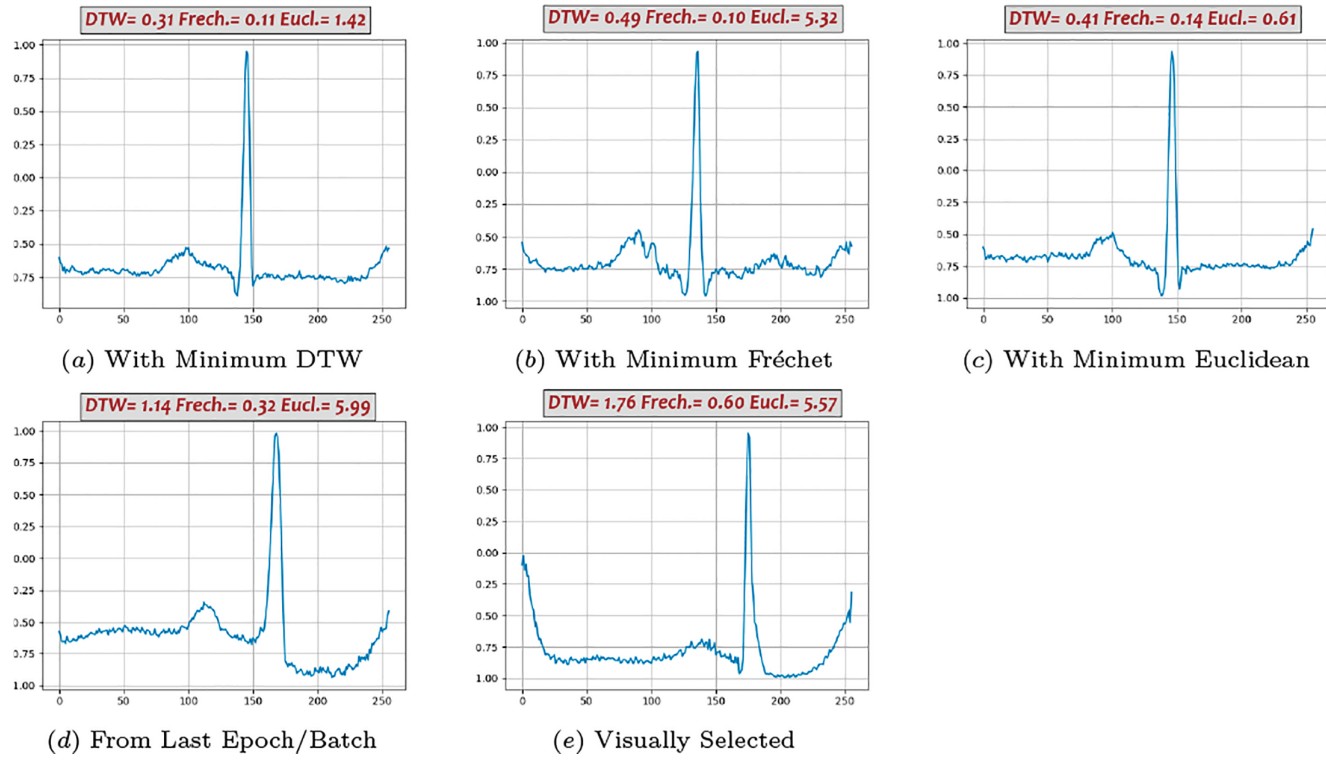

**Fig 11. Generated beats, WGAN (05).**

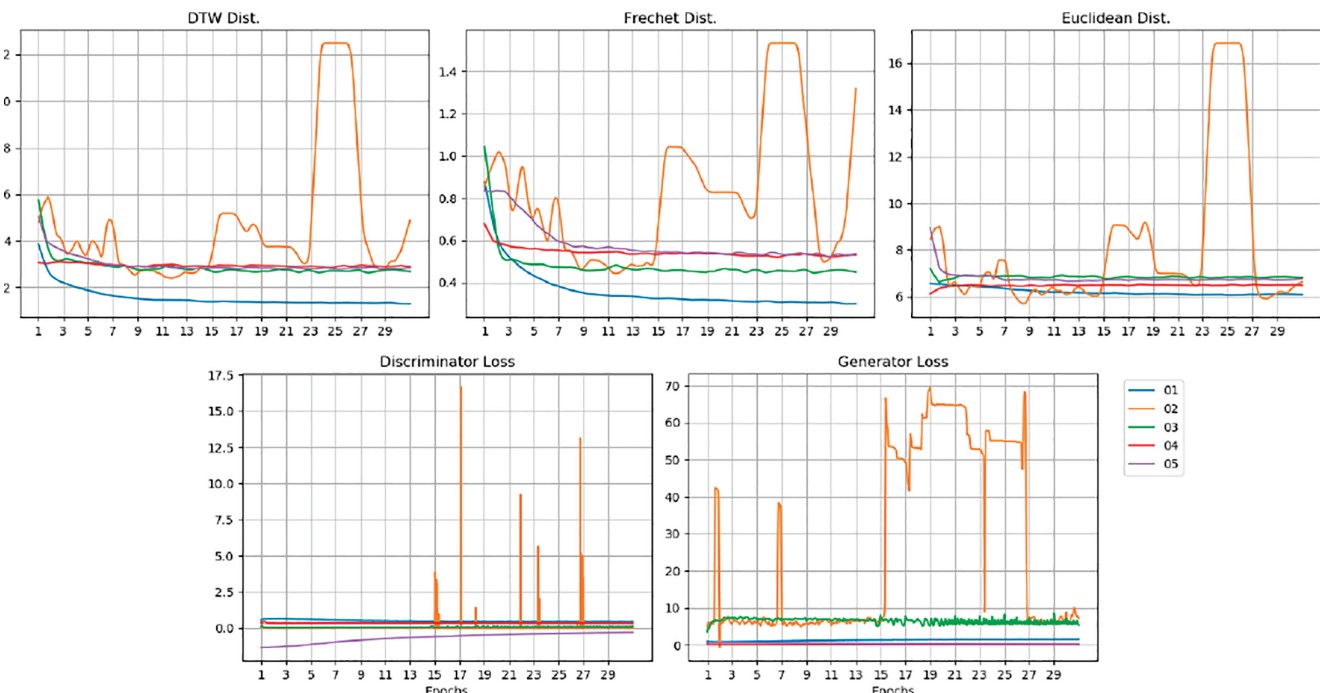

**Fig 12. Similarity measures and loss functions vs epoch numbers.**

**Table 13. Real data, balanced.**

| Cl. | Precision | Recall | F1-Score | Support |
|---|---|---|---|---|
| L | 0.95 | 0.96 | 0.96 | 1609 |
| N | 0.96 | 0.95 | 0.95 | 1607 |
| Accuracy | | | 0.95 | 3216 |
| Macro avg | 0.96 | 0.95 | 0.95 | 3216 |
| Weighted avg | 0.96 | 0.95 | 0.95 | 3216 |

**Table 14. Real data, imbalanced.**

| Cl. | Precision | Recall | F1-Score | Support |
|---|---|---|---|---|
| L | 0.52 | 1.00 | 0.68 | 1608 |
| N | 1.00 | 0.08 | 0.15 | 1608 |
| Accuracy | | | 0.54 | 3216 |
| Macro avg | 0.76 | 0.54 | 0.42 | 3216 |
| Weighted avg | 0.76 | 0.54 | 0.42 | 3216 |

**Table 15. Augmented data, balanced.**

| Cl. | Precision | Recall | F1-Score | Support |
|---|---|---|---|---|
| L | 0.99 | 0.95 | 0.97 | 1607 |
| N | 0.95 | 0.99 | 0.97 | 1609 |
| Accuracy | | | 0.97 | 3216 |
| Macro avg | 0.97 | 0.97 | 0.97 | 3216 |
| Weighted avg | 0.97 | 0.97 | 0.97 | 3216 |

**Table 16. Confusion matrices.**

**a) Real data, balanced.**

| - | L | N |
|---|---|---|
| L | 1552 | 57 |
| N | 88 | 1519 |

**b) Real data, imbalanced.**

| - | L | N |
|---|---|---|
| L | 1608 | 0 |
| N | 1480 | 128 |

**c) Augmented data, balanced.**

| - | L | N |
|---|---|---|
| L | 1521 | 86 |
| N | 9 | 1600 |

it is necessary that realistic synthetic ECG signals can be generated and made publicly available. In this study, we compared the efficiency of a few DL models in generating synthetic ECG signals using 5 different methods. The 3 introduced concepts (threshold, accepted beat and productivity rate) are employed to systematically evaluate the models. The results from Method 1 suggest that all the tested models compete very closely in generating synthetic ECG beats (Table 9). The fact that all the results are numerically in the same ballpark shows that, through this method (metric $s_1^{DF}$), all models behave more or less equally well in generating acceptable beats.

What matters in generating synthetic beats for augmenting datasets is the productivity rate ($s_4^{DF}$), i.e., the efficiency of models in terms of time and computational power, which translates into the percentage of the acceptable beats. In fact, a good model is the one that generates more *acceptable beats* per unit of time and computational power. We believe the productivity rate (Method 4) is a very efficient way to assess the capability of models in end-to-end generation of the synthetic ECG signals.

Performance analysis using Method 4 shows that Classic GAN has the highest productivity rate in terms of the DTW distance function, whereas the percentages of the BiLSTM-DC, AE/VAE-DC GAN, and WGAN models are all slightly lower but in the same ballpark, and the productivity rate of the DC-DC GAN is the lowest. Using Fréchet distance function produces the same trend, although at a slightly lower level. Thus, using Method 4, Classic GAN has the highest percentage of acceptable beats and is the most efficient model with respect to the DTW and Fréchet similarity measures. This might seem a bit counter-intuitive at first, but as FC architectures are very powerful and can potentially simulate most complicated non-linearities and functions, they can map the latent space to real data space very well. The values of the Euclidean measure are so low altogether that it does not seem to be a suitable distance function for this purpose. For instance, Fig 9(c) shows one generated beat with minimum Euclidean Distance whereas it contains none of the morphological features. The fact that both DTW and Fréchet distance functions show the same trend indicates that both are suitable for the comparison and the choice is just a matter of computational power. Visual inspection of the the generated beats (Method 5) shows that BiLSTM-DC GAN and WGAN generate acceptable beats more often than the others.

There is a lack of a systematic way for the performance assessment of models in data generating tasks, contrary to classification tasks (in which the performance metrics are standardized), and the performance is measured in practice based on the quality and quantity of the generated data on a case-based basis. We believe Methods 1 to 4 can fill the gap and provide quantitative measures for assessments of GAN family models. Our simple experiment with the state-of-the-art classifier (ECGResNet34) showed empirically that the augmentation of imbalanced ECG dataset and balancing them with synthetic ECG signals can improve the classification performance drastically.

## Future works

A better similarity measure that can capture the similarity between time series more reliably and can eliminate the supervision of humans would help greatly.

Using different loss functions in the algorithms with various regularizations that can capture the difference between time series in a better way can result in a better convergence and alleviate fluctuations in error/loss functions.

## Author contributions

**Conceptualization:** Edmond Adib.

**Data curation:** Edmond Adib.

**Formal analysis:** Edmond Adib.

**Investigation:** Edmond Adib.

**Methodology:** Edmond Adib, Fatemeh Afghah.

**Project administration:** Edmond Adib.

**Resources:** Edmond Adib.

**Software:** Edmond Adib.

**Supervision:** Edmond Adib, Fatemeh Afghah, John J. Prevost.

**Validation:** Edmond Adib, John J. Prevost.

**Visualization:** Edmond Adib.

**Writing – original draft:** Edmond Adib.

**Writing – review & editing:** Edmond Adib, Fatemeh Afghah.

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
