## [Decision Letter · Decision Letter 0]

1 Apr 2022

PONE-D-21-38425Synthetic ECG Signal Generation Using Generative Neural NetworksPLOS ONE

Dear Dr. Adib,

Thank you for submitting your manuscript to PLOS ONE. After careful consideration, we feel that it has merit but does not fully meet PLOS ONE’s publication criteria as it currently stands. Therefore, we invite you to submit a revised version of the manuscript that addresses the points raised during the review process.

please submit your revised manuscript by May 16 2022 11:59PM. If you will need more time than this to complete your revisions, please reply to this message or contact the journal office at plosone@plos.org. Please include the following items when submitting your revised manuscript:

We look forward to receiving your revised manuscript.

Kind regards,

Zahid Mehmood, PhD

Academic Editor

PLOS ONE

https://journals.plos.org/plosone/s/file?id=ba62/PLOSOne_formatting_sample_title_authors_affiliations.pdf"

“This research was partially supported by the Open Cloud Institute (OCI) at UTSA. The work of Fatemeh Afghah is supported by the National Science Foundation under Grant Number 1657260 and by the National Institute on Minority Health and Health Disparities of the National Institutes of Health under Award Number U54MD012388”

“This research was partially supported by the Open Cloud Institute (OCI) at UTSA. The work of Fatemeh Afghah is supported by the National Science Foundation under Grant Number 1657260 and by the National Institute on Minority Health and Health Disparities of the National Institutes of Health under Award Number U54MD012388.”

“FA,

National Science Foundation, Grant Number 1657260

National Institute on Minority Health and Health Disparities of the National Institutes of Health under Award Number U54MD012388

Reviewers' comments:

Reviewer's Responses to Questions

**Comments to the Author**

1. Is the manuscript technically sound, and do the data support the conclusions?

Reviewer #1: Yes

Reviewer #2: Yes

2. Has the statistical analysis been performed appropriately and rigorously? 

Reviewer #1: Yes

Reviewer #2: No

3. Have the authors made all data underlying the findings in their manuscript fully available?

Reviewer #1: Yes

Reviewer #2: Yes

4. Is the manuscript presented in an intelligible fashion and written in standard English?

Reviewer #1: Yes

Reviewer #2: Yes

5. Review Comments to the Author

Reviewer #1: This paper provides a method for ECG beat generation using Generative Neural Networks (GAN) to solve the problem of imbalanced ECG datasets. The authors developed different models (FC-FC (classic), DC-DC, BiLSTM-DC, AE/VAE-FC, and DC-DC WGAN) to find the best model that can achieves the higher performance. The authors applied their methods on the MIT-BIH dataset with five classes from it. Regarding the paper quality and structure of the paper is very well organized and clear, while the methodology is superiorly described with clear and high-resolution Figures that describe the method. Finally, the results are providing clear and discussed and compared very well. Following are my comments to authors:

1. Regarding the proposed GANs network, the author must provide details about the used training optimization method and if they can provide a comparison between different fine-tuning results using the used optimization techniques.

2. Add more Figure to show the generated ECG beats especially for the main 5 classes and labelled them.

3. Add the equations for performance evaluations.

4. I think that authors should make the data available for the public using IEEE DataPort so others can use it to evaluate their algorithms for detection of fake ECG beats.

5. Add a table the compare the proposed method with methods in and must include more recent techniques and research and organized based on the number of beats.

6. Add a plots for the developed models not only tables.

Reviewer #2: The manuscript is well organized and the idea is interesting. My main concern is that the applicability of the approach in realistic application. Specifically, I suggest to design extra experiments to compare the model evaluational performance with generated ECG sample compared to real ECG sample, for example in ECG arrhythmia classification tasks.

6. PLOS authors have the option to publish the peer review history of their article (what does this mean?). If published, this will include your full peer review and any attached files.

Reviewer #1: No

Reviewer #2: No

---

## [Author Response · Author response to Decision Letter 1]

13 May 2022

please see the attached response (rebuttal) document

---

## [Decision Letter · Decision Letter 1]

28 Jun 2022

Synthetic ECG Signal Generation Using Generative Neural Networks

PONE-D-21-38425R1

Dear Dr. Adib,

We’re pleased to inform you that your manuscript has been judged scientifically suitable for publication and will be formally accepted for publication once it meets all outstanding technical requirements.

Kind regards,

Zahid Mehmood, PhD

Academic Editor

PLOS ONE

Additional Editor Comments (optional):

Reviewers' comments:

Reviewer's Responses to Questions

**Comments to the Author**

1. If the authors have adequately addressed your comments raised in a previous round of review and you feel that this manuscript is now acceptable for publication, you may indicate that here to bypass the “Comments to the Author” section, enter your conflict of interest statement in the “Confidential to Editor” section, and submit your "Accept" recommendation.

Reviewer #1: All comments have been addressed

Reviewer #2: All comments have been addressed

2. Is the manuscript technically sound, and do the data support the conclusions?

Reviewer #1: Yes

Reviewer #2: Yes

3. Has the statistical analysis been performed appropriately and rigorously? 

Reviewer #1: Yes

Reviewer #2: Yes

4. Have the authors made all data underlying the findings in their manuscript fully available?

Reviewer #1: Yes

Reviewer #2: Yes

5. Is the manuscript presented in an intelligible fashion and written in standard English?

Reviewer #1: Yes

Reviewer #2: Yes

6. Review Comments to the Author

Reviewer #1: Dear Authors,

Thank you very much for your great response to the reviewer comments, the paper is now informative and contains all the needed information.

Reviewer #2: The main concern of the realistic use of the proposed approach has been well addressed, the author provided experiments and experimental results to validate the efficacy of augmentation.

7. PLOS authors have the option to publish the peer review history of their article (what does this mean?). If published, this will include your full peer review and any attached files.

Reviewer #1: **Yes: **Ali Mohammad Alqudah

Reviewer #2: No

---

## [Editor Report · Acceptance letter]

PONE-D-21-38425R1

Synthetic ECG signal generation using generative neural networks

Dear Dr. Adib:

I'm pleased to inform you that your manuscript has been deemed suitable for publication in PLOS ONE. Congratulations! Your manuscript is now with our production department.

Kind regards,

on behalf of

Dr. Zahid Mehmood

Academic Editor

PLOS ONE